# Thrombosis and Anemia in Pediatric Inflammatory Bowel Disease: Pathophysiology, Clinical Impact and Future Directions

**DOI:** 10.3390/ijms262110407

**Published:** 2025-10-26

**Authors:** Dragos-Florin Tesoi, Monica Hancianu, Laura Mihaela Trandafir, Manuela Ciocoiu, Maria Cristina Vladeanu, Larisa-Ioana Barbosu, Laura Bozomitu, Otilia Elena Frasinariu, Iris Bararu-Bojan, Oana-Viola Badulescu

**Affiliations:** 1Department of Pediatry, University of Medicine and Pharmacy Grigore T. Popa, 700115 Iasi, Romania; dragos-florin.tesoi@email.umfiasi.ro (D.-F.T.); laura.trandafir@umfiasi.ro (L.M.T.); barbosu.larisa-ioana@email.umfiasi.ro (L.-I.B.); laura.bozomitu@umfiasi.ro (L.B.); 2Faculty of Pharmacy, University of Medicine and Pharmacy Grigore T. Popa, 700115 Iasi, Romania; monica.hancianu@umfiasi.ro; 3Department of Pathophysiology, Faculty of Medicine, University of Medicine and Pharmacy Grigore T. Popa, 700115 Iasi, Romania; manuela.ciocoiu@umfiasi.ro (M.C.); maria.apavaloaie@umfiasi.ro (M.C.V.); iris.bararu@umfiasi.ro (I.B.-B.); oana.badulescu@umfiasi.ro (O.-V.B.)

**Keywords:** pediatric inflammatory bowel disease, anemia, thrombosis, Crohn’s disease, ulcerative colitis, inflammation, iron deficiency, hypercoagulability

## Abstract

Pediatric inflammatory bowel disease (PIBD), encompassing Crohn’s disease (CD) and ulcerative colitis (UC), is associated with inflammation that extends beyond the gastrointestinal tract. Among the most significant extraintestinal complications are anemia and thrombosis, both of which can impact disease severity, quality of life, and long-term outcomes. This review aims to explore the intertwined pathophysiology of anemia and thrombosis, clinical implications of these two complications, and management strategies for anemia and thrombosis in PIBD. Anemia is the most common systemic complication in PIBD, with multifactorial etiologies, including iron deficiency, chronic inflammation, and nutritional deficiencies. Despite its high prevalence, it remains underdiagnosed and undertreated. Thrombosis, although less frequent, poses significant risk, particularly during disease flares, hospitalizations, and in the presence of central venous catheters or corticosteroid therapy. The proinflammatory and hypercoagulable state in inflammatory bowel disease (IBD) increases thrombotic risk, necessitating early identification and, in high-risk cases, consideration of thromboprophylaxis. Anemia and thrombosis represent significant yet often overlooked complications in PIBD. Proactive screening, individualized risk stratification, and integrated management approaches are critical to improving outcomes. Further pediatric-specific research is needed to develop tailored prevention and treatment strategies.

## 1. Introduction

PIBD is a chronic, relapsing condition of the gastrointestinal tract characterized by dysregulated immune responses and mucosal inflammation. In the pediatric population, IBD is classified into three major entities: CD, UC, and indeterminate (or unclassified) colitis. Approximately 10–20% of individuals diagnosed with IBD develop the disease during childhood or adolescence, and the incidence of these pathologies among the pediatric population is continuously increasing. Beyond the primary intestinal manifestations, IBD is associated with a range of systemic complications that significantly impact morbidity and quality of life in children. Among these, thrombosis and anemia represent two of the most clinically significant extraintestinal manifestations [1,2].

Thromboembolic events (TEs) are increasingly recognized in PIBD and can be life-threatening. The pathophysiology of thrombosis in IBD is multifactorial, involving a complex interplay of chronic inflammation, endothelial dysfunction, coagulation abnormalities, and genetic predisposition [3,4,5].

Conversely, anemia is one of the most common complications of PIBD, with iron deficiency and anemia of chronic disease being the predominant subtypes. Despite its prevalence, anemia is frequently underdiagnosed or undertreated in clinical practice [6,7].

This review aims to comprehensively explore the underlying pathophysiological mechanisms, clinical impact, and future directions in the diagnosis, prevention, and management of thrombosis and anemia in PIBD. By addressing these critical aspects, we aim to enhance understanding and promote integrated, evidence-based care approaches that can improve outcomes in this vulnerable population.

## 2. Anemia in PIBD

Anemia is known to have a considerable adverse effect not only on health-related quality of life but also on academic performance and cognitive abilities, and it is a comorbid condition that is associated with other complications. Numerous studies have indicated that anemia can lead to an increased requirement for surgical interventions in individuals with IBD, as well as higher rates of hospitalization and extended lengths of hospitalization. All these elements contribute to rising healthcare expenses. Consequently, the prompt identification and suitable management of anemia related to IBD are crucial to improving the course of the disease and achieving a more favorable prognosis [7,8,9].

### 2.1. Epidemiology of Anemia in PIBD

Anemia remains one of the most frequent extraintestinal manifestations of PIBD. A 2023 multicenter study by D’Arcangelo et al. investigated children newly diagnosed with IBD and reported that approximately one-third of patients presented with anemia at diagnosis. The authors followed these children over twelve months and demonstrated partial but incomplete recovery of hemoglobin levels with oral iron supplementation, emphasizing the need for close hematologic monitoring in early disease stages [6]. More recently, a national register-based cohort study [7] reported that over one-third of children with IBD experience anemia, with a higher rate of severe anemia among those with UC. The study also noted that although awareness and treatment strategies have improved, the overall prevalence has not significantly declined in the past decade, suggesting ongoing challenges in iron status management and inflammation control. In addition, in a U.S. database study (2010–2014), anemia was observed in 51% of CD and 43% of UC pediatric patients [10]. It is important to note that the definition of anemia varied among studies, with most employing World Health Organization (WHO) reference cut-offs (hemoglobin < 11 g/dL for children under 6 years, <11.5 g/dL for those aged 6–11 years, and <12 g/dL for adolescents aged ≥ 12 years). Variations in these laboratory criteria and variable disease activity at inclusion may partially explain differences in reported prevalence rates. Patients with CD have a higher prevalence of anemia than those with UC, likely due to more pronounced systemic inflammation in CD. Additional contributing factors include vitamin B12 malabsorption from terminal ileal involvement and impaired iron absorption resulting from lesions in the proximal gastrointestinal tract [11,12,13].

### 2.2. Pathophysiology

The etiopathogenesis of IBD is intricate and influenced by multiple factors. Additionally, the anemia associated with these conditions results from multiple underlying causes. The primary mechanisms involved include iron deficiency (resulting from chronic gastrointestinal blood loss, inadequate dietary intake, or poor intestinal absorption), a persistent inflammatory state, deficiencies in essential nutrients necessary for erythropoiesis (such as vitamin B12 and folic acid), and myelosuppression caused by certain immunomodulatory treatments (including Sulfasalazine, Azathioprine, Methotrexate). Table 1 outlines the main pathophysiological mechanisms that lead to anemia in IBD, along with some important characteristics of each type of anemia. Regarding anemia associated with chronic diseases, the mechanisms that lead to the occurrence of this complication are complex and primarily based on proinflammatory cytokines, which increase hepcidin levels. Hepcidin blocks iron release from macrophages and enterocytes by targeting ferroportin, trapping iron in cells and reducing its availability for red blood cell production. Other possible causes encompass acute hemorrhage and hemolysis, which may be autoimmune, or secondary to sulfasalazine use in individuals with glucose-6-phosphate dehydrogenase (G6PD) deficiency. Recent research has identified iron deficiency as the primary cause of anemia in children newly diagnosed with IBD [11,12,14]. In the U.S. (2010–2014), 81–85% of children were screened for anemia, but only 20–24% received iron deficiency testing. Among those tested, 84% of CD patients and 86% of UC patients were iron-deficient [10].

### 2.3. Clinical Manifestations and Burden of Anemia in Pediatric Patients with IBD

The clinical presentation of anemia in patients with IBD is frequently complex and marked by a range of nonspecific symptoms. Children often report reduced energy, early exhaustion during physical activity or daily tasks, which represents one of the most common symptoms [11]. In moderate to severe anemia, fatigue can significantly impair quality of life [13]. Visible pallor of the skin and mucous membranes is also a frequent finding, particularly when anemia is severe or develops rapidly [11]. Chronic anemia in PIBD extends beyond fatigue and lowered quality of life. Accumulating evidence indicates persistent adverse effects on linear growth, pubertal progression, and neurocognitive development. Growth failure in children with IBD is multifactorial—driven by inflammation, malabsorption, dietary deficits, and corticosteroid therapy—but persistent anemia may further impair growth by limiting oxygen delivery to growth plates, reducing appetite and physical activity, and exacerbating inflammatory catabolism. Observational data show that growth deficits persist over time. In a population-based study from the Saxon pediatric IBD registry, children and adolescents demonstrated measurable growth retardation over several years [15]. Moreover, iron is critical for myelination, neurotransmitter synthesis, and energy metabolism, particularly during early childhood and adolescence when brain plasticity is high. Evidence from both general pediatric and chronic-disease populations indicates that iron deficiency—even in the absence of anemia—can impair attention, memory, and executive function, with potentially persistent effects if not corrected promptly [16,17,18]. For example, a 2025 study showed that adolescents with iron deficiency, but no anemia, had reduced basal ganglia iron content, associated with poorer cognitive performance and psychiatric symptoms [16]. Similarly, meta-analytic evidence in school-age children demonstrates that iron supplementation improves attention, memory, and intelligence scores compared with controls [17], and infant/early childhood cohorts with chronic iron deficiency exhibit lower early learning composite scores even after iron repletion [18]. In PIBD, chronic anemia has been associated with greater school absenteeism, fatigue, and diminished health-related quality of life, all of which can interfere with cognitive and psychosocial development [19,20]. Studies reporting improvement in well-being and concentration following iron repletion, underscore the functional relevance of timely diagnosis and treatment [17,18]. In PIBD, correction of iron deficiency anemia (IDA) via intravenous iron has been associated with improvements in health-related quality of life (HRQoL), including parent-reported emotional and physical domains, which likely reflect improvements in fatigue, possibly cognition [8]. Children with IBD have higher school absenteeism compared to peers without IBD. A prospective multicenter study reported that children aged 5–18 with IBD missed approximately 4.8% ± 5.5% of school days per year, versus 3.2% ± 1.6% in non-IBD controls. Many of these absences are due to scheduled medical appointments, hospitalizations, or endoscopies, but a significant proportion was directly related to disease symptoms [21]. Moderate to severe anemia reduces oxygen-carrying capacity, causing dyspnea with activities that would otherwise be tolerated [11]. Furthermore, prolonged iron deficiency—even without overt anemia—may cause symptoms such as xerostomia, cheilitis, atrophic glossitis, alopecia, or, in rarer instances, Plummer–Vinson syndrome [11].

Anemia can often go unrecognized because many of its symptoms are nonspecific. Common signs such as fatigue, pallor, decreased appetite, and poor growth can be attributed to other health issues, leading to delayed diagnosis and treatment.

### 2.4. Management Strategies

Despite a high incidence at the time of diagnosis, anemia in pediatric patients with IBD is underdiagnosed and undertreated [22]. Consensus guidelines, including the Portuguese Consensus and the SIGENP Italian PIBD working group, recommend regular monitoring of hemoglobin, ferritin, transferrin saturation, and inflammatory markers, both at diagnosis and during follow-up [23,24]. Oral iron remains a first-line therapy in many settings when iron deficiency (ID) is mild and inflammation is controlled. A 2023 study from Rome found that sucrosomial iron safely resolved anemia in ~88% of newly diagnosed PIBD patients over 12 months, with ~48% resolving within 3 months, despite variability in initial severity [6]. However, in patients with active inflammation, oral iron absorption is impaired by elevated hepcidin, and oral therapy often fails to achieve target hemoglobin increases [25,26]. Intravenous (IV) iron has become increasingly supported in PIBD for moderate-to-severe iron deficiency anemia, especially when oral iron is not tolerated, or when disease is active. In a single-center U.S. study of hospitalized PIBD patients with IDA, those treated with IV iron had a mean hemoglobin increase of ~1.9 g/dL by first ambulatory follow-up, versus ~0.8 g/dL in patients receiving oral iron or no iron [26]. Studies comparing iron sucrose and ferric carboxymaltose show that IV iron is safe and efficacious. The 2022 POPEYE randomized controlled trial compared a single dose of intravenous ferric carboxymaltose with a 12-week course of oral ferrous fumarate in children aged 8–18 years with IBD-associated anemia. The IV group showed more rapid improvement in physical fitness (6 min walk distance) at 1 month, although hemoglobin rises over 3–6 months were similar in both arms, suggesting early functional benefits of IV iron [27]. The Portuguese Consensus similarly emphasizes early active treatment of iron deficiency anemia, recommending IV iron in specific situations (when oral iron is insufficient or poorly tolerated, in active disease, or with low hemoglobin) [23]. However, guideline adherence remains suboptimal, with surveys indicating that many pediatric gastroenterologists do not routinely use IV iron even in recommended scenarios [28]. In the Italian registry, 23% of children with IBD remained anemic 1 year after diagnosis, and relapse occurs in up to ~30% during follow-up, associated with ongoing disease activity, complications, and reduced quality of life [7,29]. Oral iron side effects, particularly gastrointestinal intolerance, can limit adherence. Symptoms include nausea, epigastric discomfort, metallic taste, constipation, or diarrhea, and dark stools [6,30,31]. These adverse effects result from mucosal irritation by unabsorbed iron and hepcidin-mediated absorption alterations, often worsened by high elemental iron doses [32]. Evidence from recent trials supports alternate-day or low-dose regimens to minimize hepcidin induction and improve tolerability without compromising efficacy [30,33]. Newer formulations such as ferric maltol, sucrosomial, or liposomal iron offer improved gastrointestinal tolerability and adherence compared with traditional ferrous salts [31,32,33,34]. Practical measures include gradual dose titration, administration with small amounts of food if intolerance occurs, avoidance of calcium or antacids, and proactive management of constipation or diarrhea [6,30]. Persistent intolerance warrants switching formulations or using IV iron to ensure adherence and timely correction of iron deficiency [35,36,37].

### 2.5. The Role of Predictive Biomarkers in Diagnosis, Risk Stratification and Therapy Response

In recent years, increasing attention has been directed toward the use of specific biomarkers to enhance the diagnostic accuracy and therapeutic monitoring of anemia in PIBD. While conventional indices such as hemoglobin, ferritin, transferrin saturation, and inflammatory markers remain foundational, their interpretation is complicated by systemic inflammation. Recent pediatric and mixed-population studies indicate that biomarkers such as hepcidin, soluble transferrin receptor (sTfR), sTfR/log-ferritin index, and reticulocyte hemoglobin content can improve clinical outcomes by predicting response to iron therapy, guiding route and dosing, and identifying children at higher risk of treatment failure or recurrence.

Hepcidin reflects iron sequestration driven by inflammation, making it particularly informative in IBD. Elevated hepcidin correlates with impaired oral iron absorption. Baseline hepcidin helps predict non-responsiveness to iron therapy: higher baseline levels are associated with poor hemoglobin increase after oral iron, whereas declines following anti-inflammatory therapy often accompany hematologic recovery. Clinically relevant considerations suggest that low hepcidin favors good oral-iron absorption and trial of oral therapy, while high hepcidin indicates functional iron restriction and preferential use of IV iron or prioritizing anti-inflammatory control before oral iron [38,39].

sTfR rises with marrow iron need and is less affected by iron storage than ferritin, enabling differentiation between absolute iron deficiency and functional iron deficiency. Pediatric guidelines now endorse sTfR (and the sTfR/log-ferritin index) as a complementary tool to detect subclinical iron deficits and to identify children who are likely to respond to iron repletion despite elevated ferritin due to inflammation. In settings where ferritin is equivocal (the common scenario in active PIBD), an elevated sTfR or sTfR-index supports true iron deficiency and predicts the likelihood of hemoglobin response to iron therapy—information useful for selecting IV vs. oral iron and defining monitoring intensity [40,41].

Reticulocyte hemoglobin content (RET-He) measures the hemoglobin content of newly produced RBCs and therefore changes rapidly after iron is available to the bone marrow. In pediatric and mixed inflammatory cohorts, low baseline RET-He identifies children at high risk of progression to overt iron deficiency anemia and predicts an early hematologic response: an increase in RET-He within days–weeks after starting iron, strongly forecasts a hemoglobin rise at 4–6 weeks. Because RET-He responds quickly, it is a practical early indicator of effective iron delivery (useful after IV iron or during an oral-iron trial) and can be used to stop ineffective oral therapy early or to prompt switch to IV iron. Pediatric reference data are improving, making RET-He an increasingly reliable, rapid monitoring tool in clinical practice [42,43].

The routine use of these biomarkers in clinical practice faces numerous constraints: assay standardization (particularly for hepcidin) and pediatric reference ranges; variable influence of inflammation, infection and nutritional status on sTfR; access and cost—not all centers have hepcidin or RET-He available; a relative lack of large RCTs that randomize pediatric patients to biomarker-guided vs. standard care. Ongoing and recent observational work (including pediatric cohorts and mixed adult–pediatric studies) point toward clinical benefit from biomarker-guided strategies, but definitive outcome trials are still needed to quantify benefits on hemoglobin recovery, growth, cognition and health-care utilization [38].

In conclusion, integrating hepcidin, sTfR, and RET-He into diagnostic and monitoring pathways in PIBD improves discrimination between absolute and functional iron deficiency, predicts responsiveness to oral iron, guides dosing schedules, and enables early identification of non-responders. Implementing these biomarkers requires local validation, assay access, and clinician education, but offers a practical approach toward individualized, evidence-based anemia care in PIBD.

## 3. Thrombosis in PIBD

TE are recognized as rare but potentially life-threatening complications in pediatric patients with IBD. Although the overall incidence is lower in children compared to adults, the relative risk is significantly increased in PIBD patients compared to the general pediatric population [44,45,46,47]. Management typically involves anticoagulation therapy, usually initiated with low molecular weight heparin and followed by oral anticoagulants [46,47]. Prophylactic anticoagulation in PIBD remains controversial. Unlike adult guidelines, there is no consensus for routine thromboprophylaxis in hospitalized children with IBD, although it may be considered in selected high-risk cases [46,47]. Early identification and appropriate management of thrombotic complications can significantly reduce morbidity and improve outcomes in PIBD patients [44,45,46,47].

### 3.1. Incidence of TE in PIBD and Risk Factors

Emerging evidence underscores that pediatric patients with IBD carry a markedly elevated risk of venous thromboembolism (VTE), with incidence reaching ~31 per 10^4^ patient-years versus under 1 in unaffected children [46]. This risk is particularly high around the time of diagnosis, spanning six months before to one year after, with peak rates approximating 18 per 10^4^ person-years [44]. A French registry (EPIMAD) followed pediatric-onset IBD patients (≤16 years at diagnosis, data from 1988–2011) over a median of 8.3 years. Among 1344 patients, there were 15 VTEs (1.1%) and 2 arterial TEs (0.15%), yielding an overall thromboembolic incidence of 1.32 per 1000 person-years. Elevated risk periods included active disease, the 3-month post-surgery window, and hospitalization [48]. Among these events, cerebral venous sinus thrombosis accounts for nearly 18%, compared with ~4% in children without IBD [44]. Notably, UC appears to confer a higher VTE risk than CD, and specific factors such as hospitalization, recent surgery, steroid use, immobilization, dehydration and central venous catheter placement significantly contribute to thrombosis risk [4,46,47,49,50,51], whereas treatment with 5-aminosalicylates may offer protective benefit [48].

Prevention strategies should prioritize correction of modifiable risks during hospital stays (rehydration, early mobilization, catheter stewardship, and judicious steroid use), because evidence guiding routine pharmacologic thromboprophylaxis in hospitalized children with IBD is limited [47,49,50,52]. Research gaps remain, including precise age-specific incidence rates, the role of biologic therapies on thrombosis risk, and prospective thromboprophylaxis trials in PIBD [47,51].

### 3.2. Pathophysiological Pathways Leading to Thrombosis and Common Mechanisms That Can Cause Both Anemia and TE in PIBD

In IBD, vitamin B12 and folate deficiencies, which cause anemia and may increase thrombosis risk through elevated homocysteine (Hcy) levels, involve complex mechanisms related to impaired intestinal absorption due to inflammation or surgery, malnutrition, and medication effects—particularly sulfasalazine, which reduces folate absorption. While anemia and thrombosis share several risk factors and molecular pathways in IBD, the specific link between B12/folate deficiency and thrombosis primarily centers on their involvement in Hcy metabolism. It is now well established that elevated Hcy promotes thrombotic processes and impairs vasodilation [53,54]. Hcy exerts its effects through three primary, non-mutually exclusive mechanisms: oxidative stress induction, reduced nitric oxide (NO) bioavailability, and specific protein interactions (Figure 1). These mechanisms may act synergistically across multiple physiological levels [53,54].

Hcy participates in both auto-oxidative and oxidative reactions, either with itself or with other thiol compounds. Thiol groups (RSH), including those of Hcy, can undergo auto-oxidation in the presence of transition metals and molecular oxygen, producing reactive oxygen species (ROS) [53,55].

A second major mechanism by which Hcy exerts vascular toxicity involves diminished NO bioavailability. This phenomenon has several causes [53]:NO Trapping and Degradation: NO is rapidly inactivated by superoxide radicals via peroxynitrite formation. Elevated Hcy also inhibits glutathione peroxidase (GPx), impairing hydrogen peroxide detoxification and exacerbating oxidative stress [53].ADMA-Mediated NOS Inhibition: Hcy promotes production of asymmetric dimethylarginine (ADMA), an endogenous nitric oxide synthase (NOS) inhibitor [56].

The third major mechanism involves direct modification of vascular and plasma proteins by Hcy. According to Jacovina et al., oxidized Hcy enters endothelial cells and is reduced intracellularly, enabling it to bind to tissue plasminogen activator (tPA) through disulfide exchange with Cys9 of annexin II, thereby impairing plasminogen activation and reducing fibrinolysis [57]. Hcy also inhibits the protein C anticoagulant pathway. Activation of protein C depends on the thrombomodulin–thrombin complex, and Hcy disrupts this process by reducing disulfide bonds within thrombomodulin and protein C, impairing their anticoagulant function. Furthermore, Hcy also promotes vascular injury mediated by DNA hypomethylation. Hcy-induced hypomethylation interferes with post-translational methylation of erythrocyte membrane proteins, particularly ankyrin—a key cytoskeletal component. Defective ankyrin methylation compromises membrane integrity, reducing erythrocyte deformability and impairing microcirculatory flow [53].

Additionally, it is important to recognize the dual impact of inflammation, as it not only contributes to anemia development, but also poses a significant risk for complications such as thrombosis, a major cause of morbidity and mortality in IBD patients [58]. Inflammation and thrombosis are intricately linked under the concept of immunothrombosis, wherein immune responses activate coagulation. The interplay involves complex cellular and molecular mechanisms that converge to disrupt vascular integrity, shift the hemostatic balance toward coagulation, and suppress endogenous anticoagulant mechanisms [58].

Endothelial activation and dysfunction represent early steps in this process. Inflammatory cytokines—such as IL-1, IL-6, and TNF-α—activate endothelial cells, upregulating adhesion molecules and procoagulant factors, including tissue factor (TF) and von Willebrand factor (VWF), while downregulating anticoagulant pathways such as thrombomodulin and the protein C system. This dysfunctional endothelium loses its natural antithrombotic character and becomes a site of clot initiation [59,60].

In addition, gut barrier dysfunction and microbial translocation, such as lipopolysaccharide passage, may further activate coagulation pathway, linking gut dysbiosis to vascular risk [61].

A key component of immunothrombosis is neutrophil extracellular trap (NET) formation, or NETosis. NETs are web-like structures of chromatin, histones, and granule enzymes that trap pathogens but also promote thrombosis [62,63]. NETs serve as a scaffold for platelets, red blood cells, fibrinogen, and VWF, forming the structural backbone of clots [64,65]. NETs enhance coagulation by activating platelets, increasing thrombin generation, and supporting fibrin deposition and stability. Histones within NETs can directly activate platelets via platelet Toll-like receptors (TLR2 and TLR4), accelerating thrombin formation [64,65]. In IBD, NETs have been observed in colonic mucosa (particularly in UC), and elevated circulating NET levels likely contribute to thrombus formation [59].

Furthermore, patients with IBD frequently exhibit thrombocytosis and platelet hyperreactivity, even during quiescent disease, which is a physiological response to chronic inflammation. Elevated IL-6 stimulates hepatic production of thrombopoietin (TPO), the primary regulator of platelet production. Consequently, thrombocytosis contributes to the prothrombotic state in IBD patients [66,67].

Disease extent and complications such as fistula or stricture formation also impact TE risk [68].

### 3.3. Types of TE and Clinical Manifestations

TE can develop in a variety of vascular territories, leading to distinct clinical manifestations. The literature identifies several common thrombotic sites in PIBD, including cerebral venous sinuses, deep veins of the limbs and upper extremities (often associated with central venous catheters), abdominal vessels such as the portal and mesenteric veins, and the pulmonary circulation; less frequent sites include the retinal, intracardiac, and renal veins [44].

In a larger scoping review comprising 216 patients, cerebral venous sinus thrombosis (CVST) emerged as the most frequent form of VTE (≈34% of all VTE cases) in PIBD [4]. Although cerebrovascular events are uncommon in PIBD, approximately 3.3% of affected patients develop such complications. A retrospective analysis of 62 published cases of cerebrovascular events in children with IBD examined risk factors, clinical characteristics, affected vascular territories, and outcomes. Most of the cerebrovascular events occurred when IBD was in an active phase—nearly 88% of the children experienced a flare at the time of the event. On average, the cerebrovascular event occurred approximately 20.8 weeks after IBD onset. Persistent and severe headache represented the most frequent presenting symptom, followed by seizures, motor or sensory deficits, vomiting, altered consciousness, and visual disturbances [69,70]. CVST is most commonly involved the transverse sinuses, followed by the superior sagittal and sigmoid sinuses. In cases of cerebral arterial infarction (CAI), the middle cerebral artery (MCA)—particularly the right MCA—was most frequently affected [69,70].

Deep vein thrombosis (DVT), particularly involving the lower limbs, is also observed in hospitalized PIBD patients and is associated with risk factors such as central venous catheter use and systemic corticosteroid therapy [47]. The clinical consequences of DVT include not only the local symptoms—pain, swelling, impaired mobility—but also serious complications such as pulmonary embolism (PE), which may present with sudden dyspnea, chest pain, or tachypnea. Another significant concern is post-thrombotic syndrome (PTS). According to a meta-analysis of 12 studies including more than 1100 pediatric patients, approximately 25–30% of children who develop DVT subsequently develop PTS [71].

Splanchnic vein thrombosis, including portal vein thrombosis (PVT), has also been documented in PIBD and can exacerbate abdominal symptoms, complicating diagnosis. Clinical manifestations of PVT include abdominal pain, hepatosplenomegaly, gastrointestinal bleeding, and ascites—many of which arise secondary to portal hypertension. Several case reports have been described PVT as the initial presentation of CD, suggesting that thrombosis may precede or unmask IBD in some patients [72,73].

VTE contributes substantially to morbidity, mortality, and healthcare costs in PIBD. Hospitalized children with IBD who develop VTE experience significantly longer hospital stays (median 11 vs. 5 days) and have markedly higher rates of intensive care unit (ICU) admission (30.2% vs. 4.8%). In-hospital mortality is also elevated (1.5% vs. 0.2% in those without VTE) [74]. Although most children achieve full recovery, mortality rates of up to 5% have been reported in some series [4]. Beyond clinical outcomes, the burden of VTE encompasses psychological stress for families, the necessity for long-term anticoagulation monitoring, and potential chronic sequelae.

### 3.4. Management

Current recommendations, including those from the European Society for Paediatric Gastroenterology, Hepatology and Nutrition (ESPGHAN) and the European Crohn’s and Colitis Organisation (ECCO) guidelines for the management of acute severe colitis (ASC) in children, advise the use of low-molecular-weight heparin (LMWH) prophylaxis in pubertal children with ASC who possess at least one additional risk factor for thrombosis (e.g., obesity, smoking, use of oral contraceptives, complete immobilization, central venous catheterization, concurrent significant infection, known prothrombotic disorder, previous VTE, or family history of VTE). In prepubertal children, prophylaxis is recommended only when two or more of these risk factors are present [75]. Conversely, consensus statements released by the Canadian Association of Gastroenterology explicitly advise against the routine use of venous thromboembolism prophylaxis in pediatric patients hospitalized for IBD, even when admission is due to a severe disease flare [76]. In 2022, an international RAND-based panel evaluated the appropriateness of thromboprophylaxis in hospitalized children experiencing IBD flares. The panel’s findings suggested that thromboprophylaxis may be appropriate in a broader range of clinical scenarios than those currently specified in existing guidelines, indicating the need for potential guideline updates. Scenarios deemed “appropriate” include: all children with new-onset ASC, all exacerbations of known UC—even in the absence of additional risk factors (except for prepubertal patients with limited disease and no risk factors)—and children with CD and concurrent risk factors during hospitalization [77].

A recent guideline development document [78] provides further evidence-based recommendations for pediatric IBD management. Prophylactic anticoagulant therapy is recommended for hospitalized children with severe IBD and a history of thrombosis, with LMWH as the preferred agent. Therapeutic anticoagulation is indicated for children with IBD who have developed TE. LMWH is generally the first-line option for acute management in hospitalized children due to its predictable pharmacokinetics and reduced need for laboratory monitoring compared with unfractionated heparin. Fondaparinux is an alternative, particularly in patients who cannot tolerate heparin or in those with heparin-induced thrombocytopenia. Direct oral anticoagulants (DOACs) are increasingly used in pediatric VTE; however, data in IBD remain limited. Caution is warranted due to potential issues with gastrointestinal absorption, elevated bleeding risk during active disease, drug interactions, and the limited evidence base in younger age groups [78].

For the first episode of VTE in a child with clinically inactive IBD (in remission) and resolved or reversible risk factors, anticoagulation for a minimum of three months is recommended, provided that risk factors have remained absent for at least one month before discontinuation [78]. If the child experiences VTE during active disease, anticoagulation should be continued until three months after clinical remission is achieved [78]. For recurrent VTE, or in the presence of persistent risk factors (e.g., ongoing active disease, CVCs, immobilization, or thrombophilia), extended anticoagulation therapy may be considered [79].

Some centers have adopted standardized management algorithms. For example, Nationwide Children’s Hospital implemented a protocol in 2023 for PIBD admissions. According to this protocol, patients with moderate-to-severe colitis (or severe ileitis with an additional risk factor) receive enoxaparin prophylaxis. Patients who do not meet these criteria are managed with mechanical prophylaxis (sequential pneumatic compression devices, graduated compression stockings) or non-pharmacologic measures such as mobilization and hydration. Implementation of this algorithm increased appropriate prophylaxis rates to approximately 93.8% of admissions.

Despite these advances, randomized controlled trials assessing thromboprophylaxis versus no prophylaxis in PIBD are still lacking, and the optimal balance between bleeding and thrombosis risk remains uncertain. Existing evidence is largely derived from observational studies, which, although informative, possess inherent methodological limitations.

### 3.5. Current Biomarkers for Diagnosing Thrombosis and Monitoring Therapeutic Response in PIBD

In PIBD, there is growing interest in identifying and validating biomarkers not only for disease activity but also for thrombotic risk and treatment response. In general, pediatric VTE populations, several hemostatic parameters—such as D-dimer, fibrinogen, platelet count, white blood cell count, and factor VIII—have been repeatedly investigated. In a recent systematic review of over 10^4^ children, these markers emerged as the most promising predictors of VTE risk, although their sensitivity and specificity remain variable across studies [80]. In patients with IBD, elevated D-dimer levels and fibrin/fibrinogen degradation products (FDPs) have been detected during active disease, with higher levels correlating with increased disease severity in UC [81,82]. Regarding treatment response, biologic therapy—particularly anti-tumor necrosis factor (anti-TNF) agents—appears to ameliorate the procoagulant imbalance. For example, studies demonstrate that anti-TNF therapy reduces elevated factor VIII concentrations, improves the factor VIII/protein C ratio, and decreases thrombin generation capacity in IBD patients [83]. In the future, such coagulation-related parameters may serve as surrogate biomarkers to evaluate therapeutic efficacy of anti-TNF agents and to estimate thrombotic risk reduction in IBD patients.

Nevertheless, there remains a paucity of literature specifically addressing biomarkers validated for diagnosing thrombosis in IBD, as opposed to general VTE risk or disease activity. Many existing studies are cross-sectional, include small sample sizes, or fail to adequately control for confounding variables such as inflammation, catheter use, or corticosteroid exposure. Well-designed prospective studies measuring baseline thrombophilic markers, tracking their longitudinal changes during therapy, and correlating them with clinical thrombotic outcomes, are needed to establish reliable biomarkers for both diagnosis and monitoring of therapeutic response in PIBD.

## 4. Interplay Between Anemia and Thrombosis

Anemia and thrombosis are complex and clinically significant extraintestinal manifestations of IBD, with emerging evidence indicating that their coexistence is not coincidental but mechanistically interconnected. In patients with IBD, chronic inflammation acts as a shared pathogenic driver of both anemia and thrombotic risk, fostering a systemic proinflammatory and prothrombotic state.

### 4.1. Role of Anemia in Increasing Thrombosis Risk

Although anemia is commonly recognized for its consequences such as fatigue, impaired oxygen delivery, and compensatory cardiovascular responses, accumulating evidence over recent years indicates that specific forms of anemia are associated with an increased risk of both arterial and venous thrombotic events. The mechanisms are multifactorial, involving direct effects of RBC abnormalities and hemolysis, alterations in coagulation, rheological changes, platelet dysfunction, endothelial injury, and iron deficiency-mediated pathways, among others.

Hemolytic anemias, although rare in IBD, provide clear examples of anemia promoting thrombosis. Intravascular hemolysis—whether immune-mediated (e.g., autoimmune hemolytic anemia) or drug-induced (e.g., sulfasalazine)—triggers multiple prothrombotic processes. RBC lysis releases cell-free hemoglobin, heme, and free heme iron, which exert several downstream effects. Free hemoglobin scavenges NO, a key vasodilator and inhibitor of platelet and endothelial activation. Reduced NO bioavailability leads to vasoconstriction, enhanced platelet activation and increased leukocyte-endothelial and platelet–endothelial adhesion [84]. Additionally, arginase released from lysed RBCs depletes L-arginine—the substrate for NO synthase—further impairing NO bioavailability [84]. Hemolysis also induces oxidative stress through the release of pro-oxidant molecules, generating excessive ROS that damage endothelial cells and activate coagulation cascades. ROS further exacerbate NO depletion by forming peroxynitrite, a cytotoxic oxidant species [84,85]. Moreover, RBC injury or membrane stress promotes the release of microvesicles (MVs) enriched in phosphatidylserine (PS), which provide a negatively charged surface facilitating activation of the intrinsic tenase complex (factors VIII and IX). These RBC-derived hemolytic microvesicles (HMVs) have been shown to enhance thrombin generation in plasma and propagate coagulation [84,85]. Under physiological conditions, PS is confined to the inner leaflet of the RBC membrane, but in hemolytic or oxidative stress states, PS externalization occurs, supporting assembly of procoagulant complexes and promoting RBC-endothelial adhesion [84].

In addition to hemolytic mechanisms, IDA—the most prevalent anemia subtype in PIBD—also contributes to thrombosis risk. In IDA, the bone marrow responds with reactive thrombocytosis and heightened platelet reactivity, elevating thrombotic potential [86,87]. For example, Aslam et al. (2022) reported a case of recurrent arterial thrombosis (affecting the spleen, kidney, and abdominal aorta) in a patient with severe IDA and thrombocytosis; correction of anemia normalized platelet counts and prevented further events [88]. Similarly, iron deficiency is an underrecognized risk factor for recurrent VTE, including PE and DVT, predominantly mediated via thrombocytosis [89]. Recent study (2020–2021) demonstrated that iron deficiency and combined conditions (such as administration of estrogen) upregulate transferrin expression, which interacts with coagulation and anticoagulant factors to promote hypercoagulability. Specifically, transferrin enhances the activity of thrombin and Factor XIIa, while blockade of transferrin-thrombin and transferrin-Factor XIIa interactions reduces thrombosis in experimental models [90].

Furthermore, increased endogenous erythropoietin (EPO) production—or administration of exogenous erythropoiesis-stimulating agents (ESAs)—may indirectly contribute to thrombosis by stimulating platelet production, modulating endothelial function, and augmenting thrombin generation. While meta-analyses in surgical populations have not consistently shown increased mortality or major thrombosis with ESA use compared to iron therapy alone, the potential for increased VTE risk warrants caution [91,92].

Anemia also induces compensatory hemodynamic and rheological alterations. To offset reduced oxygen-carrying capacity, cardiac output rises, and blood viscosity decreases due to lower RBC mass. These changes increase blood flow velocity and reduce laminar stability, predisposing to turbulent flow, particularly in regions of vascular branching or narrowing. Turbulence and shear stress fluctuations can damage endothelial cells, promoting inflammatory activation, platelet aggregation, and coagulation initiation [87,93,94].

Finally, chronic tissue hypoxia resulting from anemia further amplifies ROS production, endothelial dysfunction, and procoagulant surface expression. For instance, a 2023 murine study demonstrated that repeated blood loss induces endothelial injury through oxidative stress [94,95]. Janaszak-Jasiecka et al. similarly reported that hypoxia impairs NO synthesis, increases ROS generation, and precipitates vascular endothelial dysfunction—a central mechanism linking anemia to thrombosis [93]. All mechanisms by which anemia promotes thrombosis in IBD are summarized in Table 2.

### 4.2. Mechanisms by Which Thrombosis May Worsen Anemia in Patients with IBD

Given the multiple mechanisms through which anemia contributes to thrombotic risk in individuals with IBD, it is equally important to recognize that thrombosis can, in turn, exacerbate anemia in this population. Microvascular obstruction reduces local tissue perfusion and oxygenation, thereby aggravating mucosal injury and blood loss, which further adds to IDA secondary to chronic gastrointestinal bleeding. Localized ischemia and the formation of microthrombi within the intestinal mucosa amplify inflammatory injury and hemolysis. Microvascular infarctions have been documented in IBD—even in the absence of overt inflammation—suggesting that thrombosis may act both as a cause and a consequence of mucosal damage [96,97]. Moreover, interactions between activated platelets and endothelial cells play a pivotal role in perpetuating inflammation. Upon platelet activation, particularly through CD40L signaling, platelets engage endothelial and immune cells, stimulating the release of proinflammatory cytokines. This cytokine surge upregulates hepcidin, the regulator of iron metabolism, thereby impairing intestinal iron absorption and promoting anemia of inflammation [97,98]. Conversely, therapeutic anticoagulation used for TE or prophylaxis can increase gastrointestinal bleeding risk, thereby worsening anemia in patients with IBD. A 2024 meta-analysis demonstrated that while prophylactic anticoagulation during hospitalization significantly reduced the incidence of VTE, it concurrently elevated the risk of major bleeding, a trade-off that may adversely influence anemia outcomes [99]. Therefore, thrombosis in IBD should not be viewed solely as an extraintestinal manifestation but also as a potential driver of anemia progression—particularly during active disease phases when inflammatory and coagulative pathways are upregulated. [58,100].

The interplay between thrombosis and anemia in IBD forms a bidirectional and self-perpetuating pathogenic loop. Thrombosis exacerbates anemia through ischemia, inflammatory activation, and blood loss, while anemia promotes thrombosis via hypoxia-induced endothelial dysfunction and platelet hyperreactivity. Therapeutic strategies aimed at modulating inflammation, improving iron homeostasis, and mitigating thrombotic risk, may therefore disrupt this vicious cycle and improve overall clinical outcomes in PIBD populations.

## 5. Discussion

To ensure comprehensive and up-to-date coverage, we performed a narrative literature review using PubMed, Scopus, and Web of Science. We prioritized articles published within the past seven years (2019–2025) but included earlier landmark studies when they provided foundational or pediatric-specific data. Search terms combined “pediatric inflammatory bowel disease”, “anemia”, “iron deficiency”, “thrombosis”, and “venous thromboembolism”. Eligible studies included observational cohort studies, randomized controlled trials, systematic reviews, meta-analyses, and consensus guidelines. Pediatric cohorts (≤18 years) were prioritized, although mixed-age studies were considered if pediatric-specific data could be extracted. Case reports and small case series were included selectively to illustrate rare but clinically relevant presentations, such as cerebral venous sinus thrombosis. Reference lists of key papers were also screened to identify additional relevant studies.

Experimental and imaging data [16,18] link iron deficiency to altered brain iron content and cognitive performance, supporting the earlier hypothesis that chronic anemia contributes to growth retardation and neurodevelopmental deficits in PIBD. However, few longitudinal pediatric studies quantify these long-term sequelae, representing a notable evidence gap. Most investigations focus on hematologic correction rather than functional recovery, limiting the understanding of broader developmental consequences.

Therapeutic evidence in pediatric cohorts favors IV iron for moderate-to-severe iron deficiency anemia or in the presence of active inflammation. Comparative studies [26,27] demonstrate faster hemoglobin recovery and earlier improvement in physical fitness with IV ferric carboxymaltose versus oral iron, consistent with impaired oral absorption in high-hepcidin states. However, long-term hemoglobin outcomes appear similar once inflammation subsides, underscoring that disease control remains the primary determinant of hematologic recovery. The Portuguese and SIGENP guidelines [23,24] accordingly advocate individualized routes based on disease activity and tolerance. Nonetheless, surveys [28] reveal low adherence to IV protocols, reflecting cost, access, and safety concerns. Oral iron innovations—sucrosomial, ferric maltol, or liposomal formulations—achieve better tolerability, but pediatric comparative data are scarce and often single-center.

Recent work on hepcidin, sTfR, and RET-He highlights their potential to refine diagnosis and predict iron therapy response. High hepcidin levels correlate with poor oral iron absorption, whereas early increases in RET-He forecast hematologic improvement. Despite these promising markers, lack of assay standardization and pediatric reference ranges currently limit clinical application. Incorporating such biomarkers into routine algorithms could promote personalized management of anemia in PIBD.

Management of thrombosis in PIBD is constrained by limited pediatric trial data. Consensus discrepancies persist: ESPGHAN-ECCO guidelines recommend low-molecular-weight heparin prophylaxis for adolescents with acute severe colitis and risk factors (2021–2024), whereas the Canadian Association of Gastroenterology discourages routine prophylaxis. The RAND panel (2022) sought to harmonize these positions, expanding prophylaxis indications but acknowledging insufficient evidence on bleeding risk. Implementation studies such as the Nationwide Children’s Hospital protocol (2023) demonstrate that algorithmic risk assessment increases adherence without apparent harm, yet randomized controlled trials remain absent. Consequently, extrapolation from adult experience dominates current practice.

Biomarker research for thrombotic risk is even more incipient than for anemia. Elevated D-dimer and factor VIII levels correlate with disease activity rather than specific thrombotic outcomes, reducing specificity. Small studies suggest anti-TNF therapy may normalize coagulation profiles, indirectly reducing risk, but prospective validation is lacking.

The article’s novelty lies in its integrated approach to two interconnected but often separately studied complications of PIBD: anemia and thrombosis. While most previous research has focused on these complications of IBD individually, this review is distinctive because it explores the shared pathophysiological mechanisms linking anemia and thrombosis in PIBD. In addition, it emphasizes the bidirectional relationship between these complications, illustrating how anemia may contribute to thrombotic risk and, conversely, how thrombosis can exacerbate anemia, highlighting the intertwined clinical and biological pathways that worsen disease outcomes. Ultimately, our review advocates for a holistic management approach, suggesting that addressing one complication, such as correcting anemia, may also reduce thrombotic risk and improve overall prognosis.

Regarding future medical directions, the article outlines several forward-looking strategies and research priorities:-development of pediatric-specific diagnostic and therapeutic algorithms for both anemia and thrombosis, as adult guidelines are not directly applicable to the pediatric population;-exploration of biomarkers—including hepcidin, soluble transferrin receptor, and reticulocyte hemoglobin content—to improve diagnostic accuracy and facilitate early detection of iron metabolism disturbances and thrombotic risk;-conducting prospective longitudinal and randomized clinical studies to determine the safety and efficacy of iron supplementation and anticoagulation therapies in children with IBD;-assessing the long-term impact of chronic anemia on growth, cognitive function, and school performance in pediatric populations;-implementing integrated, multidisciplinary care strategies to enhance screening, prevention, and management of extraintestinal complications in PIBD.

## 6. Conclusions

This review challenges the traditional view of treating anemia and thrombosis as separate complications, emphasizing the need for integrated clinical management, and it identifies shared pathophysiological pathways (inflammation, immunothrombosis, oxidative stress, cytokine activity) that can be therapeutic targets. It also highlights the lack of pediatric-specific research and guidelines, urging future clinical trials and biomarker studies. Therefore, future research should prioritize multicenter pediatric trials evaluating biomarker-guided iron therapy, standardized thromboprophylaxis protocols, and long-term neurodevelopmental outcomes of chronic anemia. Integrating hematologic and coagulation monitoring into PIBD management pathways may mitigate cumulative morbidity and improve quality of life.

## Figures and Tables

**Figure 1 ijms-26-10407-f001:**
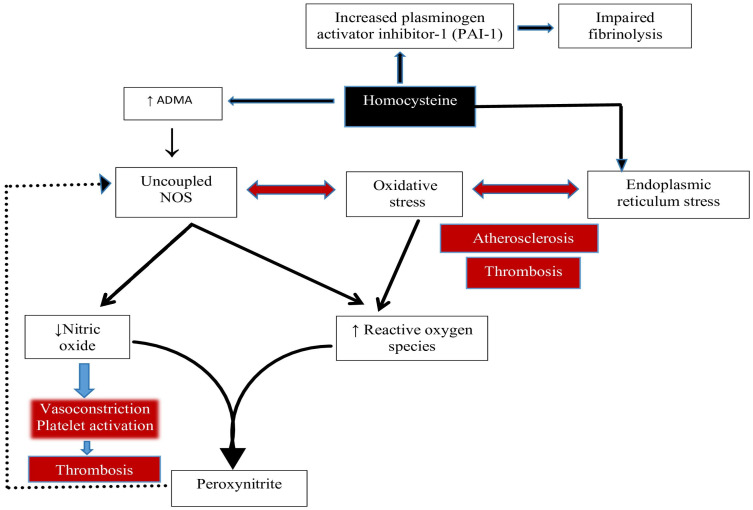
Oxidative stress caused by increased levels of Hcy promotes endothelial dysfunction and thrombosis through reactive oxygen species. Moreover, homocysteine increases asymmetric dimethylarginine (ADMA) levels, which may uncouple nitric oxide synthase (NOS), reducing nitric oxide (NO) production and increasing superoxide levels. The reaction between superoxide and NO forms peroxynitrite, decreasing NO availability and contributing to endothelial dysfunction. Additionally, peroxynitrite may further uncouple NOS. Hcy can also interfere with the body’s ability to dissolve clots by increasing plasminogen activator inhibitor-1 levels (PAI-1). PAI-1 inhibits fibrinolysis by reducing plasmin formation.

**Table 1 ijms-26-10407-t001:** Mechanisms of Anemia in PIBD.

Mechanism	Pathophysiology	Clinical Relevance/Notes [12,14]
Iron deficiency anemia	Chronic blood loss from inflamed gastrointestinal mucosa, reduced iron absorption due to inflammation (especially in CD affecting the duodenum)	The most common cause; it may coexist with anemia of chronic disease.
Anemia of chronic disease	Inflammation → ↑ Hepcidin → ↓ Iron release from stores and ↓ absorption	normocytic or microcytic anemia; poor response to oral iron.
Vitamin B12 deficiency	Terminal ileum involvement or resection impairs B12 absorption	macrocytic anemia; especially in CD with ileal disease/resection.
Folate deficiency	Malabsorption, poor intake, or medication-related (methotrexate, sulfasalazine)	macrocytic anemia; less common than B12 deficiency.
Acute post-hemorrhagic anemia	Anemia caused by sudden and significant blood loss	accompanied by severe bleeding manifested by symptoms such as melena, hematemesis, rectal bleeding
Bone marrow suppression	Inflammatory cytokines suppress erythropoiesis (e.g., IL-6, TNF-α).	It can result in pancytopenia or isolated anemia; associated with low reticulocyte count.
Medication-induced anemia	Sulfasalazine, azathioprine, and methotrexate can impair hematopoiesis.	It requires monitoring of the complete blood count (CBC) regularly.
Hemolysis	Autoimmune hemolytic anemia or secondary to drugs (rare in IBD)	Consider in cases with elevated lactate dehydrogenase (LDH), reticulocytes, and low haptoglobin; Coombs-positive anemia possible (especially in UC)

**Table 2 ijms-26-10407-t002:** Mechanisms by which anemia in inflammatory bowel disease can promote thrombosis.

Mechanism	Description
Endothelial dysfunction	Chronic anemia and hypoxia damage endothelium, promoting a prothrombotic state.
Increased platelet reactivity	Certain anemia types cause platelet activation or enhanced aggregation.
Elevated EPO	High EPO levels stimulate platelet production and activation.
Turbulent blood flow	Reduced hematocrit increases shear stress and turbulence, promoting endothelial activation.
Free hemoglobin/heme	In hemolysis, free hemoglobin scavenges NO, inducing vasoconstriction and platelet activation.
Microparticle release	RBC- and platelet-derived microparticles exhibit strong procoagulant activity.
Iron deficiency-induced thrombocytosis	Reactive thrombocytosis in IDA increases thrombotic risk.

Abbreviations: EPO, erythropoietin; NO, nitric oxide; RBCs, red blood cells.

## Data Availability

No new data were created or analyzed in this study. Data sharing is not applicable to this article.

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
