# Peer review of "Thrombosis and Anemia in Pediatric Inflammatory Bowel Disease: Pathophysiology, Clinical Impact and Future Directions"

_ijms, 2025, doi:10.3390/ijms262110407_

Round 1

Reviewer 1 Report

Comments and Suggestions for Authors

The manuscript represents a valid and well-structured contribution in the field of IBD The authors addressed a complex and highly clinically relevant topic such as anemia and thrombosis in pediatric patients with inflammatory bowel disease, managing to clearly integrate the data. Persè's text does not need to be changed or modified, there are some spelling errors that need to be fixed:
1. Some names are not written for long and the acronym is used directly (example PIBD which is used without specifying what is meant)
2. Some references are placed after the period and others before, they are usually placed before the sentence ends with the period.
3. Recheck the text thoroughly, such as ''supression'' which should be ''suppression'
4. In fact, mixed terminology is sometimes used “thromboembolism” instead of thromboembolism, correct
5. Some passages (eg on the prevalence of anaemia or the risk of thrombosis in paediatric IBD) are repeated in several sections with almost the same sentences.
6. “Thrombosis, particularly venous thromboembolism (VTE), is a less frequent but serious complication...” appears twice in chapter 5
Thoroughly recheck the text and writing in English to allow the reader maximum understanding of the text.

Apart from the spelling issues, I would recommend that the authors address the following points.

  1. Inclusion of more robust quantitative data, such as meta-analyses or relevant clinical studies, maybe the most recent work from 2024/25.
  2. Clarifying the methodology of literature selection for the references cited. For example, specifying whether the authors prioritized studies published in the past two years, included only pediatric cohorts, or considered certain study designs. Providing more detail on how the evidence was gathered would strengthen the rigor and transparency of the manuscript.

  3. A deeper discussion on the long-term consequences of chronic anemia in pediatric IBD, particularly regarding growth, neurocognitive development, which heavily effect the child growths.
  4. Exploration of predictive biomarkers (e.g., hepcidin, soluble transferrin receptor, reticulocyte hemoglobin content) not only for diagnosis but also for therapy response and risk stratification.
  5. Adding a dedicated section on currently used biomarkers, both for the diagnosis of anemia and thrombosis in pediatric IBD and for confirming treatment response. This would enhance the clinical relevance of the review and provide readers with practical insights.

In conclusion, while I consider the manuscript well-written and highly informative, addressing these points could further enhance its scientific value and clinical applicability.

Author Response

Dear Reviewer,

We would like to sincerely thank you for the time and effort you have invested in reviewing our manuscript. We greatly appreciate your insightful and constructive comments, which have helped us to improve the quality and clarity of our work.

Please find below our point-by-point responses to your observations. The changes have been incorporated into the revised manuscript, and all modified sections are marked accordingly.

Comments:

The manuscript represents a valid and well-structured contribution in the field of IBD The authors addressed a complex and highly clinically relevant topic such as anemia and thrombosis in pediatric patients with inflammatory bowel disease, managing to clearly integrate the data. Persè's text does not need to be changed or modified, there are some spelling errors that need to be fixed:

  1. Some names are not written for long and the acronym is used directly (example PIBD which is used without specifying what is meant)
  2. Some references are placed after the period and others before, they are usually placed before the sentence ends with the period.
  3. Recheck the text thoroughly, such as ''supression'' which should be ''suppression'
  4. In fact, mixed terminology is sometimes used “thromboembolism” instead of thromboembolism, correct
  5. Some passages (eg on the prevalence of anaemia or the risk of thrombosis in paediatric IBD) are repeated in several sections with almost the same sentences.
  6. “Thrombosis, particularly venous thromboembolism (VTE), is a less frequent but serious complication...” appears twice in chapter 5

Thoroughly recheck the text and writing in English to allow the reader maximum understanding of the text.

Response:

We have carefully reviewed the entire manuscript for spelling, grammar, and consistency. Specific corrections include:

- Expanded all acronyms upon first mention (e.g., pediatric inflammatory bowel disease [PIBD]).

- Standardized the placement of references before punctuation marks.

- Corrected typographical errors such as “supression” → “suppression.”

- Ensured consistent use of terminology, e.g., replaced all instances of “thromboembolism” with the standardized term venous thromboembolism (VTE).

- Removed repeated phrases, including duplicate sentences such as “Thrombosis, particularly venous thromboembolism (VTE), is a less frequent but serious complication…” (Chapter 5).

  1. Inclusion of more robust quantitative data, such as meta-analyses or relevant clinical studies, maybe the most recent work from 2024/25.

            Response: We have updated the manuscript to include data from recent meta-analyses and clinical studies. These additions strengthen the evidence base regarding the prevalence and risk factors of anemia and thrombosis in pediatric IBD. The newly added references are marked in red in the revised reference list.

  1. Clarifying the methodology of literature selection for the references cited. For example, specifying whether the authors prioritized studies published in the past two years, included only pediatric cohorts, or considered certain study designs. Providing more detail on how the evidence was gathered would strengthen the rigor and transparency of the manuscript.

Response: A detailed description of the literature selection process has been added to the Discussion section. Priority was given to publications from the last seven years (2019–2025).

  1. A deeper discussion on the long-term consequences of chronic anemia in pediatric IBD, particularly regarding growth, neurocognitive development, which heavily effect the child growths.

Response: We have added a new passage in „2.3. Clinical manifestations and burden of anemia in pediatric patients with IBDsubsection, discussing the long-term impact of chronic anemia on growth, neurocognitive development, and overall quality of life in pediatric IBD patients. This section emphasizes how chronic anemia can negatively affect physical growth, cognitive outcomes, and psychosocial functioning.

  1. Exploration of predictive biomarkers (e.g., hepcidin, soluble transferrin receptor, reticulocyte hemoglobin content) not only for diagnosis but also for therapy response and risk stratification.

Response: We have expanded the discussion on emerging biomarkers, including hepcidin, soluble transferrin receptor (sTfR), and reticulocyte hemoglobin content (CHr). The revised text now discusses their diagnostic utility, role in predicting therapy response, and potential in risk stratification for in pediatric IBD.

  1. Adding a dedicated section on currently used biomarkers, both for the diagnosis of anemia and thrombosis in pediatric IBD and for confirming treatment response. This would enhance the clinical relevance of the review and provide readers with practical insights.

            Response: Two new subsections titled „2.5. The role of predictive biomarkers in diagnosis, risk stratification and therapy response” and „3.4. Current Biomarkers for Diagnosing Thrombosis and Monitoring Therapeutic Response in PIBD” have been added. This addition enhances the clinical applicability of the review.

We thank you once again for your thoughtful comments and suggestions. We believe that the revisions made in response to your feedback have significantly strengthened the manuscript. We hope that the updated version meets your expectations and the standards of the journal.

Sincerely,
Dr. Dragos-Florin Tesoi,  on behalf of all authors

Reviewer 2 Report

Comments and Suggestions for Authors

Dear Authors,

Thank you for submitting your manuscript to the International Journal of Molecular Sciences. The topic, anemia and thrombosis in pediatric inflammatory bowel disease, is clinically important and timely. However, after reviewing the manuscript carefully, I believe that it requires major revision before it can be considered for publication. Below are my detailed comments:

  1. While the review is thorough, much of the content is descriptive and repeats well-known epidemiology and textbook physiology. The manuscript does not clearly define what new synthesis or unique conceptual framework it provides. Please clarify how your review advances knowledge beyond existing literature.

  2. Several parts (particularly Introduction, Anemia sections 2.1-2.3, and Thrombosis sections 3.1-3.3) are extremely long and dense. The text could be shortened and refocused to highlight the most clinically relevant points and emerging evidence rather than repeating basic pathophysiology in detail.

  3. Most cited data are presented uncritically. There is little discussion of study quality, limitations of available pediatric data, or controversies (e.g., guidelines for thromboprophylaxis). Adding critical commentary and gaps would make the review stronger.

  4. The manuscript repeatedly explains similar mechanisms (e.g., iron deficiency pathways, thrombosis risk factors) in multiple places. Streamlining would improve readability.

  5. The current conclusion is too generic and does not synthesize the main messages or offer actionable insights for clinicians or researchers. Strengthen this section by providing a clear, critical summary and future research directions.

  6. Some citations are outdated or not the most authoritative. Please update with the most recent, high-impact pediatric IBD studies and guidelines.

  7. The text would benefit from careful editing for grammar, clarity, and conciseness. 

  8. Some subsections (e.g., extensive molecular details about homocysteine pathways, oxidative stress) seem too deep for the review’s clinical focus and dilute the practical message. Consider condensing or moving to supplemental material.

  9. The review does not sufficiently translate mechanistic detail into practical recommendations (screening frequency, risk stratification, or management strategies specific to pediatric care). Strengthen clinical applicability.

Author Response

Dear Reviewer,

We would like to sincerely thank you for the time and effort you have invested in reviewing our manuscript. We greatly appreciate your insightful and constructive comments, which have helped us to improve the quality and clarity of our work.

Please find below our point-by-point responses to your observations. The changes have been incorporated into the revised manuscript, and all modified sections are marked accordingly.

Comments:

  1. While the review is thorough, much of the content is descriptive and repeats well-known epidemiology and textbook physiology. The manuscript does not clearly define what new synthesis or unique conceptual framework it provides. Please clarify how your review advances knowledge beyond existing literature.

            Response: We agree that the initial version was largely descriptive. The revised manuscript now clearly defines its conceptual contribution in Discussion section. Specifically, we emphasize our novel approach, which integrates hematologic and thrombotic complications as interconnected outcomes of chronic inflammation and systemic dysregulation in pediatric IBD.

  1. Several parts (particularly Introduction, Anemia sections 2.1-2.3, and Thrombosis sections 3.1-3.3) are extremely long and dense. The text could be shortened and refocused to highlight the most clinically relevant points and emerging evidence rather than repeating basic pathophysiology in detail.

Response: We have shortened and reorganized the Introduction and Sections 2.1–2.3 (Anemia) and 3.1–3.3 (Thrombosis). Redundant descriptions of basic physiology were condensed, and repetitive information was removed.

  1. Most cited data are presented uncritically. There is little discussion of study quality, limitations of available pediatric data, or controversies (e.g., guidelines for thromboprophylaxis). Adding critical commentary and gaps would make the review stronger.

           Response: We have revised the text throughout to include a more analytical perspective, especially in Discussion section.

  1. The manuscript repeatedly explains similar mechanisms (e.g., iron deficiency pathways, thrombosis risk factors) in multiple places. Streamlining would improve readability.

            Response: We carefully reviewed all sections for redundancy and merged overlapping content.

  1. The current conclusion is too generic and does not synthesize the main messages or offer actionable insights for clinicians or researchers. Strengthen this section by providing a clear, critical summary and future research directions.

            Response: The Conclusion has been completely rewritten. It now provides a concise synthesis of the key findings and specific recommendations for future research, including prospective studies on early biomarker-based risk stratification, individualized thromboprophylaxis, and long-term outcome monitoring.

  1. Some citations are outdated or not the most authoritative. Please update with the most recent, high-impact pediatric IBD studies and guidelines.

            Response: All references have been thoroughly reviewed and updated. We have replaced older citations with the most recent and high-impact pediatric IBD studies, including those published in 2019–2025. We have retained only a limited number of older references because their scientific content is still pertinent to the context of our article.

  1. The text would benefit from careful editing for grammar, clarity, and conciseness.

            Response: The entire manuscript has undergone comprehensive English language editing for grammar, clarity, and conciseness. Long sentences were simplified, and technical terms were standardized. We hope the revised version reads more fluently and is now more accessible to a broad clinical and academic audience.

  1. Some subsections (e.g., extensive molecular details about homocysteine pathways, oxidative stress) seem too deep for the review’s clinical focus and dilute the practical message. Consider condensing or moving to supplemental material.

            Response: We have shortened sections that contained extensive molecular detail (e.g., homocysteine metabolism, oxidative stress). Only mechanisms directly relevant to clinical outcomes are retained.

  1. The review does not sufficiently translate mechanistic detail into practical recommendations (screening frequency, risk stratification, or management strategies specific to pediatric care). Strengthen clinical applicability.

Response: We have expanded several sections to translate mechanistic insights into practice-oriented recommendations. This includes, for example, encouraging screening frequency for anemia and thrombosis in pediatric IBD, or proposed use of specific biomarkers for early detection and treatment monitoring.

We thank you once again for your thoughtful comments and suggestions. We believe that the revisions made in response to your feedback have significantly strengthened the manuscript. We hope that the updated version meets your expectations and the standards of the journal.

Sincerely,
Dr. Dragos-Florin Tesoi,  on behalf of all authors

Reviewer 3 Report

Comments and Suggestions for Authors

ABSTRACT
I would like to thank the authors for exploring such an important and relevant topic in the field of pediatric IBD. In the first lines of the abstract, you use the terms “systemic” and “beyond the gastrointestinal tract,” which, in the context of IBD, may sound somewhat repetitive. I would recommend, for example, removing “systemic” and keeping the reference to inflammation extending beyond the gastrointestinal tract to improve the flow and readability of the text.

INTRODUCTION
Thank you for introducing the topic and clearly elucidating the concept of IBD, especially in the pediatric context. I noticed that some words contain an unnecessary dash (e.g., child-hood, pop-ulation, predis-positions…). I would suggest writing these terms in their correct form.

2.1
I appreciated the introduction and the clear presentation of anemia as one of the most prevalent complications in pediatric IBD. I would suggest using the appropriate abbreviations directly (such as IBD), as they have already been defined earlier in the text, without necessarily repeating the full term.

Thank you for presenting data from several relevant studies. I would suggest specifying the laboratory criteria or cut-off values used to define anemia in these data.

2.2
As aforementioned I would suggest using the appropriate abbreviations directly (such as IBD), as they have already been defined earlier in the text, without necessarily repeating the full term.

I would like to thank the authors for clearly presenting the main etiopathogenetic mechanisms of anemia in pediatric IBD. However, including both “persistent inflammatory state” and “proinflammatory cytokines” might sound somewhat repetitive, as they refer to closely related concepts. I would suggest combining them into a single, more concise statement.

Table 1
Thank you for exploring the main mechanisms of anemia. I would recommend adding the appropriate references to support the information reported in Table 1, particularly in the section “Clinical relevance/notes.”

2.3
Thank you for presenting the clinical manifestations and consequences of anemia in children. I would recommend specifying what IDA stands for, as it is not defined earlier in the text.

2.4
Thank you for clearly describing the management of anemia in IBD patients and for explaining the differences between oral and intravenous iron supplementation.
Considering that clinical practice is not always straightforward and that it is sometimes difficult to ensure the preferred route of administration (for example, due to a patient’s or parent’s refusal of IV iron therapy even in cases of moderate-to-severe disease activity), I would suggest including a brief explanation of the main symptoms of oral iron intolerance and possible strategies to prevent or manage them, in order to improve patients’ adherence to therapy.

3.
As aforementioned I would suggest using the appropriate abbreviations directly (such as IBD), as they have already been defined earlier in the text, without necessarily repeating the full term.

I would like to thank the authors for presenting the main aspects of thromboembolic events in pediatric IBD, including pathophysiology, risk factors, and management.
I would suggest distributing the references throughout the paragraph rather than placing them all at the end. Since different concepts are presented, linking each statement to its specific source would improve clarity and strengthen the scientific reliability of the section.

3.1
Thank you for evaluating the incidence of VTE in pediatric IBD. Since the abbreviation “PIBD” has not been defined previously, I would suggest spelling out the full term before using the abbreviation.

3.2
As aforementioned I would suggest using the appropriate abbreviations directly (such as IBD), as they have already been defined earlier in the text, without necessarily repeating the full term.

Thank you for explaining the pathophysiological pathways involved in thrombosis in IBD patients. I would suggest using the abbreviation for homocysteine (Hcy), as introduced earlier, after first specifying it in the text.

I would like to thank the authors for clearly explaining the metabolic pathways involved in homocysteine metabolism. I would suggest slightly refining the sentence “via the remethylation pathway (requires folate and vitamin B12) or transsulfuration pathway (requires vitamin B6)” for better readability and flow. Using the form “requiring” instead of “requires” could improve sentence fluidity and align the style with the rest of the paragraph.

I appreciated the detailed explanation of the molecular mechanisms by which homocysteine affects fibrinolysis. However, the cited author appears to be incorrect — the study cited describing the interaction between homocysteine, annexin II, and tPA was conducted by Jacovina et al., not Jacobsen. I would recommend correcting the reference accordingly and using the appropriate “et al.” format when referring to multiple authors.

As aforementioned, I would recommend correcting references using the appropriate “et al.” format when referring to multiple authors (e.g. “Jamaluddin et al”).

Thank you for providing a detailed description of the fibrinolytic changes and additional risk factors in IBD. I would suggest that abbreviations such as tissue plasminogen activator (t-PA) and thromboembolic events (TE) and others, which have already been introduced earlier in the manuscript, could be used directly here without repeating the full terms. This would help streamline the text and improve readability.

3.3
As aforementioned I would suggest using the appropriate abbreviations directly (such as IBD), as they have already been defined earlier in the text, without necessarily repeating the full term.

I would suggest defining the abbreviation “PVT” at its first appearance, as introducing an unexplained abbreviation may reduce clarity for readers unfamiliar with the term.

I would suggest using the appropriate abbreviations directly (such as VTE), as they have already been defined earlier in the text, without necessarily repeating the full term (“Venous thromboembolism (VTE) in hospitalized children with IBD is linked to longer hospital stays (median 11 vs. 5 days), [...]).

I would suggest defining the abbreviation “ICU” at its first appearance, as introducing an unexplained abbreviation may reduce clarity for readers unfamiliar with the term.

3.4
I would suggest using the appropriate abbreviations directly (such as LMWH), as they have already been defined earlier in the text, without necessarily repeating the full term.

4.
As aforementioned I would suggest using the appropriate abbreviations directly (such as IBD), as they have already been defined earlier in the text, without necessarily repeating the full term.

I would like to thank the authors for describing the proposed mechanisms linking anemia and thrombosis. I would suggest that abbreviations such as reactive oxygen species (ROS) and nitric oxide (NO), which have already been introduced earlier in the manuscript, could be used directly here without repeating the full terms. This would help streamline the text and improve readability.

I would suggest defining the abbreviation “ESA” at its first appearance, as introducing an unexplained abbreviation may reduce clarity for readers unfamiliar with the term.

Table 2
Thank you for summarizing in a table the main mechanisms by which anemia in inflammatory bowel disease can promote thrombosis — this effectively enhances the clarity and impact of the proposed mechanisms. I would suggest including, below the table, a note defining all abbreviations used. Although these have already been explained in the main text, tables should be self-contained and understandable independently, in accordance with scientific reporting standards.

4.2
I would suggest using the appropriate abbreviations directly (such as VTE), as they have already been defined earlier in the text, without necessarily repeating the full term.

5.
I would suggest using the appropriate abbreviations directly (such as IBD and VTE), as they have already been defined earlier in the text, without necessarily repeating the full term.

Author Response

Dear Reviewer,

We would like to sincerely thank you for the time and effort you have invested in reviewing our manuscript. We greatly appreciate your insightful and constructive comments, which have helped us to improve the quality and clarity of our work.

Please find below our point-by-point responses to your observations. The changes have been incorporated into the revised manuscript, and all modified sections are marked accordingly.

Comments:

ABSTRACT

I would like to thank the authors for exploring such an important and relevant topic in the field of pediatric IBD. In the first lines of the abstract, you use the terms “systemic” and “beyond the gastrointestinal tract,” which, in the context of IBD, may sound somewhat repetitive. I would recommend, for example, removing “systemic” and keeping the reference to inflammation extending beyond the gastrointestinal tract to improve the flow and readability of the text.

Response: The term “systemic” has been removed, and the sentence now refers only to inflammation extending beyond the gastrointestinal tract to improve clarity and flow.

INTRODUCTION

Thank you for introducing the topic and clearly elucidating the concept of IBD, especially in the pediatric context. I noticed that some words contain an unnecessary dash (e.g., child-hood, pop-ulation, predis-positions…). I would suggest writing these terms in their correct form.

Response: We appreciate this observation. All typographical inconsistencies have been corrected (e.g., “childhood,” “population,” “predispositions”).

2.1

 I appreciated the introduction and the clear presentation of anemia as one of the most prevalent complications in pediatric IBD. I would suggest using the appropriate abbreviations directly (such as IBD), as they have already been defined earlier in the text, without necessarily repeating the full term. Thank you for presenting data from several relevant studies. I would suggest specifying the laboratory criteria or cut-off values used to define anemia in these data.

Response: Thank you for this suggestion. The abbreviation IBD is now used directly throughout the section. Additionally, we have specified the laboratory cut-off values used to define anemia, as reported in the cited studies.

2.2

 As aforementioned I would suggest using the appropriate abbreviations directly (such as IBD), as they have already been defined earlier in the text, without necessarily repeating the full term. I would like to thank the authors for clearly presenting the main etiopathogenetic mechanisms of anemia in pediatric IBD. However, including both “persistent inflammatory state” and “proinflammatory cytokines” might sound somewhat repetitive, as they refer to closely related concepts. I would suggest combining them into a single, more concise statement.

Response: We agree with this helpful recommendation. The abbreviation IBD is used directly, and the statements on inflammation have been merged into a single, concise sentence to reduce redundancy.

Table 1

Thank you for exploring the main mechanisms of anemia. I would recommend adding the appropriate references to support the information reported in Table 1, particularly in the section “Clinical relevance/notes.”

Response: We appreciate this comment. Appropriate references have been added to Table 1 to support the reported information.

2.3

Thank you for presenting the clinical manifestations and consequences of anemia in children. I would recommend specifying what IDA stands for, as it is not defined earlier in the text.

Response: We thank the reviewer for this observation. The term IDA (iron deficiency anemia) is now defined at its first appearance.

2.4

Thank you for clearly describing the management of anemia in IBD patients and for explaining the differences between oral and intravenous iron supplementation. Considering that clinical practice is not always straightforward and that it is sometimes difficult to ensure the preferred route of administration (for example, due to a patient’s or parent’s refusal of IV iron therapy even in cases of moderate-to-severe disease activity), I would suggest including a brief explanation of the main symptoms of oral iron intolerance and possible strategies to prevent or manage them, in order to improve patients’ adherence to therapy.

Response: Thank you for this valuable suggestion. A short paragraph has been added describing the main symptoms of oral iron intolerance and strategies to improve adherence and minimize adverse effects.

3.

 As aforementioned I would suggest using the appropriate abbreviations directly (such as IBD), as they have already been defined earlier in the text, without necessarily repeating the full term. I would like to thank the authors for presenting the main aspects of thromboembolic events in pediatric IBD, including pathophysiology, risk factors, and management. I would suggest distributing the references throughout the paragraph rather than placing them all at the end. Since different concepts are presented, linking each statement to its specific source would improve clarity and strengthen the scientific reliability of the section.

Response: We appreciate this comment. Abbreviations are now used consistently, and references have been distributed throughout the section to link each statement with its respective source.

3.1

Thank you for evaluating the incidence of VTE in pediatric IBD. Since the abbreviation “PIBD” has not been defined previously, I would suggest spelling out the full term before using the abbreviation.

Response: Thank you for pointing this out. Pediatric inflammatory bowel disease (PIBD) is now defined at its first mention.

3.2

 As aforementioned I would suggest using the appropriate abbreviations directly (such as IBD), as they have already been defined earlier in the text, without necessarily repeating the full term.

Thank you for explaining the pathophysiological pathways involved in thrombosis in IBD patients. I would suggest using the abbreviation for homocysteine (Hcy), as introduced earlier, after first specifying it in the text.

I would like to thank the authors for clearly explaining the metabolic pathways involved in homocysteine metabolism. I would suggest slightly refining the sentence “via the remethylation pathway (requires folate and vitamin B12) or transsulfuration pathway (requires vitamin B6)” for better readability and flow. Using the form “requiring” instead of “requires” could improve sentence fluidity and align the style with the rest of the paragraph.

I appreciated the detailed explanation of the molecular mechanisms by which homocysteine affects fibrinolysis. However, the cited author appears to be incorrect — the study cited describing the interaction between homocysteine, annexin II, and tPA was conducted by Jacovina et al., not Jacobsen. I would recommend correcting the reference accordingly and using the appropriate “et al.” format when referring to multiple authors.

As aforementioned, I would recommend correcting references using the appropriate “et al.” format when referring to multiple authors (e.g. “Jamaluddin et al”).

Thank you for providing a detailed description of the fibrinolytic changes and additional risk factors in IBD. I would suggest that abbreviations such as tissue plasminogen activator (t-PA) and thromboembolic events (TE) and others, which have already been introduced earlier in the manuscript, could be used directly here without repeating the full terms. This would help streamline the text and improve readability.

Response: We have implemented all suggested changes. Abbreviations such as Hcy are used consistently, the sentence describing the remethylation and transsulfuration pathways has been completely eliminated, and the reference has been corrected to Jacovina et al. The “et al.” format has also been applied consistently throughout.

3.3

As aforementioned I would suggest using the appropriate abbreviations directly (such as IBD), as they have already been defined earlier in the text, without necessarily repeating the full term. I would suggest defining the abbreviation “PVT” at its first appearance, as introducing an unexplained abbreviation may reduce clarity for readers unfamiliar with the term. I would suggest using the appropriate abbreviations directly (such as VTE), as they have already been defined earlier in the text, without necessarily repeating the full term (“Venous thromboembolism (VTE) in hospitalized children with IBD is linked to longer hospital stays (median 11 vs. 5 days), [...]). I would suggest defining the abbreviation “ICU” at its first appearance, as introducing an unexplained abbreviation may reduce clarity for readers unfamiliar with the term.

Response: We thank the reviewer for this clarification. The abbreviations portal vein thrombosis (PVT) and intensive care unit (ICU) are now defined at first mention, and abbreviations are used consistently thereafter.

3.4

I would suggest using the appropriate abbreviations directly (such as LMWH), as they have already been defined earlier in the text, without necessarily repeating the full term.

Response: Thank you for the observation. The abbreviation LMWH is now used directly throughout the text.

4.

As aforementioned I would suggest using the appropriate abbreviations directly (such as IBD), as they have already been defined earlier in the text, without necessarily repeating the full term. I would like to thank the authors for describing the proposed mechanisms linking anemia and thrombosis. I would suggest that abbreviations such as reactive oxygen species (ROS) and nitric oxide (NO), which have already been introduced earlier in the manuscript, could be used directly here without repeating the full terms. This would help streamline the text and improve readability. I would suggest defining the abbreviation “ESA” at its first appearance, as introducing an unexplained abbreviation may reduce clarity for readers unfamiliar with the term.

Response: All abbreviations are now used consistently, and ESA (erythropoiesis-stimulating agents) is defined at first mention.

Table 2

 Thank you for summarizing in a table the main mechanisms by which anemia in inflammatory bowel disease can promote thrombosis — this effectively enhances the clarity and impact of the proposed mechanisms. I would suggest including, below the table, a note defining all abbreviations used. Although these have already been explained in the main text, tables should be self-contained and understandable independently, in accordance with scientific reporting standards.

Response: A note defining all abbreviations has been added below Table 2 to make it self-contained.

4.2

I would suggest using the appropriate abbreviations directly (such as VTE), as they have already been defined earlier in the text, without necessarily repeating the full term.

Response: We thank the reviewer for this comment. Abbreviations such as VTE are now used directly throughout the section.

5.

 I would suggest using the appropriate abbreviations directly (such as IBD and VTE), as they have already been defined earlier in the text, without necessarily repeating the full term.

Response: Abbreviations are now used consistently in this section and throughout the manuscript.

We thank you once again for your thoughtful comments and suggestions. We believe that the revisions made in response to your feedback have significantly strengthened the manuscript. We hope that the updated version meets your expectations and the standards of the journal.

Sincerely,
Dr. Dragos-Florin Tesoi,  on behalf of all authors

Reviewer 4 Report

Comments and Suggestions for Authors

I can't recall the last time I encountered a paper that so comprehensively addressed the pathophysiology of certain disorders. The presentation of the etiopathogenesis of anemia and thrombosis in pediatric patients with IBD, along with their interrelations, is exceptionally detailed. Congratulations to the authors.

Author Response

Dear Reviewer,

We would like to express our sincere gratitude for your thoughtful and encouraging comments regarding our manuscript. Your generous words - especially your appreciation of the comprehensive discussion of the pathophysiology and the detailed presentation of the etiopathogenesis of anemia and thrombosis in pediatric patients with IBD - mean a great deal to us.

We are truly honored by your positive evaluation and grateful that our efforts to present this complex topic with clarity and depth have been recognized.

Your feedback provides us with great motivation to continue our research in this field.

With our deepest appreciation,

Associated Professor Oana-Viola Badulescu, on behalf of all authors